# The Relationship of Chilean Minors with Brands and Influencers on Social Networks

**Beatriz Feijoo** [1,*] and **Charo Sádaba** [2]

1   Department of Communication, School of Business and Communication, International University of La Rioja (UNIR), 26006 Logroño, Spain

2   School of Communication, University of Navarra, 31080 Pamplona, Spain; csadaba@unav.es

*   Correspondence: beatriz.feijoo@unir.net

**Abstract:** This article presents the results of a study that sought to analyze the relationship between minors and brands on social media. The frequency with which minors search for or share information or subscribe to brand web pages was measured, as well as their following of influencers, who commonly refer to consumer goods. The main purpose of this article is to contribute to learning about the commercial environment that surrounds children in their routines on social media, particularly because of their growing influence in home purchasing decisions. The results, obtained from a survey applied in 501 homes in the Metropolitan Area of Santiago de Chile to minors between 10 and 14 years old, show that the respondents effectively interact with brands through social media. Although it is not a widespread practice among 10- to 12-year-olds, it is increasingly becoming present among 13- to 14-year-olds. Children seem most interested in sportswear, fashion, and technology brands, areas in which children have significant influence in family purchasing decision. Following influencers through social media is also a common activity among minors. In particular, the age groups here studied preferred to follow celebrities, particularly from the worlds of music, football, or YouTube, over specific brands.

**Keywords:** children; brands; social networks; consumer; influencers; commercial content





## 1. Introduction

The relevance of this study lies in the increasing presence of electronic commerce worldwide–Chile included—and the ever-growing possibilities brands have to access customers, particularly children who are totally immersed in the digital environment, and thus are becoming direct consumers or agents with the capacity to influence family purchasing decisions [1]. According to the Report Kaspersky Lab 2018–2019 an increasing percentage of minors visit online shopping sites; in 2019, 14.18% of visits to major e-commerce platforms were performed by children, an increase from the 2.83% in 2018 [2].

It is important to note that members of the youngest age range included in this study (ages 10 to 14) have already developed a consumer role, and thus are current as well as future consumers, and therefore a target audience for many consumer goods brands [3]. At the same time, this age group may not yet have the personal resources necessary to critically and freely deal with commercial content due to their age and maturity level. In this line of research, this study seeks to contribute to understanding how children interact with brands and influencers on social media in terms of frequency.

## 2. The Relationship of Minors with Commercial Content in the Digital Context

Children's exposure to commercial content and its impact have been investigated in recent years, and particularly with regard to the advertising of certain products, such as food, alcohol, and tobacco. However, the relationship of minors with commercial brands and its possible effect on children's behavior as current and future consumers has received less attention [4].

Minors in consumer societies grow up surrounded by brands as the latter are part of children's environment [5,6]. According to John, by the age of 3, children are already able to recognize trademarks [7]. This familiarity can contribute to the nurturing and growth of brand awareness, brand image, and brand knowledge, vital dimensions for building brand equity, and highly valued for the medium and long-term strategic vision of brands [8]. Additionally, brand recognition or recall is affected by certain age aspects related to children's evolutionary development. The recalling process, unlike recognition, requires two cognitive steps: the mental search for a certain memory unit and the verification of whether that unit is correct or not, an ability child usually develop from the age of 7 on [9]. Between the ages of 4 and 8, the speed of information processing increases rapidly [10,11]. As children grow, so does their knowledge and thus they can relate brands they do not recognize with knowledge previously stored in their memory which helps stimulate their memory [11]. Longitudinal studies such as those conducted by Guest [12] have revealed that the contact that children have with brand is reflected in consumption patterns as adults, although the opposite may also be true, that is, early childhood familiarity with brands may not determine purchase decisions upon reaching adulthood [5].

In the academic discussion on whether children's early age links with brands go beyond recognition, Ji proposes a clear criterion to demonstrate the existence of such a bond: "when children retain memories of their interaction with brands" [5] (p. 372). With regards to this statement, it could be argued that the strategies implemented by companies to reach children in the digital environment, which use hybrid formats and generate positive experiences with a high social component, can more easily contribute to the creation of these memories [13,14]. In addition, the degree of intimacy with which some of these devices are used and content is accessed, particularly in the case of smartphones, could also make it easier for these experiences to leave a deeper imprint on users and lead to reinforcing the bond with brands [15].

In the digital realm, companies seek out children by using attractive formulas, which include ways to involve them through games and entertainment and by emotionally empathizing with them through the interactivity and connectivity that derives from relating with their contacts on social networks. As Nairn and Fine rightly point out, many of these formats involve affective associations, rather than objective or rational messages, which can render the persuasive knowledge of minors ineffective [16]. The ethical implications derived from this reality must be taken into account [17], particularly when the children themselves point out this commercial content as something annoying in the digital environment [18]. The establishment of agreements with influencers is also a recent and popular way to reach children on these platforms [19].

According to Lou and Yuan, a social media influencer is a "content generator who has a status of expertise in a specific area and who has cultivated a sizable number of captive followers on social media by regularly producing valuable content" [20] (p. 59). One of their most significant characteristics is the credibility they achieve with their followers, with whom they create an intimate, quasi-familiar atmosphere [21]. Once this positive relationship is established, they may be able to persuade their audience to have certain opinions or attitudes or even modify their audience's behavior toward a phenomenon, product, or service [22].

Various studies have analyzed the reasons that may contribute to developing influencers' credibility. Xiao, Wang, and Chan-Olmsted identified trustworthiness, social influence, argument quality, and information involvement as key factors [23]. Lou and Yuan consider that the credibility of influencers is built on their range of experience, trustworthiness, their attractiveness and the ability to resemble their audience [20]. Studies have shown that endorsement by expert influencers is more successful than that by celebrities when promoting certain products [24]. The levels of credibility and admiration influencers have achieved are reflected in a significant increase in positive participation regarding products and services to which they make reference [25,26].

It has become evident that the routines and digital preferences of 10- to 14-year-olds, a group Bringué and Sádaba have termed "the interactive generation" have evolved. However, as these authors also stated, this cohort is still characterized by the use of high-level technological equipment, as well as being permanently connected, multitasking, and technologically-speaking, precocious, and emancipated [27]. Minors' preference for mobile phones has become a consolidated fact over recent years, as the penetration and use of this device continually grows among children, both in Chile [28–30] and globally [31].

Chile is an interesting case study due to its access to and consumption of the internet through mobile devices [32]. Some 85% of the population is connected to the internet, within the range of other OECD (Organisation for Economic Co-operation and Development) countries [29].

Internet is mostly accessed from a mobile device (84.2%) [29], and 92% of children and adolescents in Chile have a smartphone [28]. Within the OECD range of amount of time connected to internet after school, Chilean children are connected the longest; while the OECD average is of 130 min per week, in Chile that figure reaches almost 200 min [33]. Despite socioeconomic differences in terms of the equipment used and location (urban or rural), smartphone penetration is socially uniform [34].

Changes in the routines of use is twofold: the consolidation of mobile phones as the preferred display screen, and the integration of peer relationships and other practices such as playing online video games or participating in social media [27].

These arguments support the need for developing new approaches and conducting in-depth studies to learn about the context in which minors relate to brands, either directly or through influencers.

## 3. Materials and Methods

The objective of this study is to determine the frequency with which children interact with brands, products and influencers through social media and analyze their use of these platforms as 10- to 14-year-old consumers.

To reach the proposed objective, the following research questions were formulated:

- RQ 1—How often do children search for information, share content, or follow brand and influencer profiles on social media?
- RQ 2—Are these routines of use subject to significant differences depending on the children's age, gender, and socioeconomic level?
- RQ 3—Which brands and influencers present on social media elicit the most interest among the minors surveyed?

The results presented in this article belong to a larger project funded by the National Commission for Scientific and Technological Research of the Government of Chile (Fondecyt Initiation N° 11170336) on children, advertising and mobile devices. This study applied quantitative methodology supported by face-to-face surveys distributed throughout households in the Metropolitan Area of Santiago, Chile with children ages 10 to 14 years with the aim of learning about how minors the use and consumption that minors make of mobile phones early on and then deepen their relationship with the commercial content they receive through this device. A sample of 501 households was defined and one child per residence was surveyed (once profile criteria had been met).

According to the 2017 Chilean Census, the Metropolitan Area has a total of 373,129 residences with children ages 10 to 14. Locally, it is standard to divide the area geographically into sectors or macrozones, and thus households were located as follows: Center = 47,148; North = 50,553; East = 69,954; West = 73,877; South = 131,597.

100 cases were assigned to each macrozone and the sample was distributed proportionally based on the percentage of households with children ages 10 to 14 in each commune (local administrative division) that conform the 5 macrozones. The result was a probabilistic design by areas (macrozones) with an error of $\pm$ 4.4% under the assumptions of simple random sampling and 95% confidence. The field work took place between the months of May and July 2018.

The fact that for methodological reasons minors directly participate in the fieldwork is in itself a though-provoking point of this project. Its goal is to seek to increase our knowledge on the audience here described and to examine its relationship with mobile phones and advertising, while respecting and protecting children's rights at all times. This article aims to contribute to a better understanding of the role children have in the mechanisms of media and advertising, considering at all times their developmental state and moment in their education and striving to respect and defend their dignity and rights as a human beings. To safeguard the integrity of participants and researchers prior to collecting the data here analyzed, written authorization was requested from the guardian who signed an informed consent and the minors themselves were also asked for their assent. These documents were reviewed and validated by the Ethics Committee of the university to which this research project is linked (Universidad de Los Andes, Santiago, Chile).

A social studies company was in charge of the field work (Feedback S.L) who constructed their network of interviewers with previous experience in research studies with minors available to the authors. Households in which a minor aged 10 to 14 lived were randomly selected within each commune in each macrozone. In those cases in which there was more than one individual who met the selection characteristics, the one who had his birthday closest to the day of the survey was selected. Firstly, the interviewer explained the essence of the research to the parent or adult responsible for the household, who had to issue a signed consent for the minor to participate in the study. Next, the assent of the minor himself also had to be obtained. Finally, in a neutral area of the home (kitchen or living room) the questionnaire was completed with a maximum duration of 20–25 min through an electronic device, with the aim of conserving the child's attention. In total, 501 valid responses were obtained from the minors surveyed. The main characteristics of the sample are summarized in the Table 1:

**Table 1.** Main characteristics of the sample.

| | |
|---|---|
| Age | 10–12: 60% (300)<br>13–14: 40% (201) |
| Minor's education | 5th grade: 24% (120)<br>6th grade: 24% (121)<br>7th grade: 20% (100)<br>8th grade: 16% (80)<br>1st high school: 16% (80) |
| Gender | Men: 46% (230)<br>Women: 54% (271) |
| N° of members per household | 2 members: 4.8% (24)<br>3 members: 15.8% (79)<br>4 members: 32.7% (164)<br>5 members: 24.2% (121)<br>6 members: 11.8% (59)<br>7 or + members: 8.8% (44)<br>No answer: 2% (10) |
| Household socioeconomic status | High (C1): 7.2% (36)<br>Middle (C2 and C3): 46.9% (235)<br>Low (D): 42.9% (215)<br>No answer: 3% (15) |
| Mobile technology equipment | Smartphones: 99% (496)<br>Notebook/laptop: 52% (261)<br>Tablet: 49% (245) |
| Main uses for mobile technology | Entertainment/leisure: 83% (416)<br>Communication: 77% (386)<br>Check/be active on social networks: 43% (215) |

Compiled by the authors.

This research project stems from the results obtained in thematic area of the questionnaire dedicated to social media. For this part of the study, a five-point Likert scale was used to determine the frequency with which children perform certain activities on social media (Table 2). Frequency was coded as follows for analysis by SPSS: never (1); infrequently (2); somewhat frequently (3); frequently (4); and very frequently (5). Specifically, children were asked about the following routines of use:

- Contacting: contacting close friends or acquaintances.
- Searching for info and sharing content: searching for information on current events; sharing information on current events; uploading personal content/comments.
- Engaging with brands and influencers: searching for information on products, brands, sales; sharing information on products, brands and sales; subscribing to influencers' fan page; subscribing to brand fan pages; Participating in promotional contests.

**Table 2.** Statistical descriptors of the elements used in the Likert scale.

| Routine | Descriptors | Minors ($N = 501$) ($\alpha = 0.811$) | | | | |
| --- | --- | --- | --- | --- | --- | --- |
| | | Mín. | Máx. | Mean | $S^2$ | SD. |
| Contacting | Contact friends/acquaintances | 1 | 5 | 3.46 | 1.713 | 1.309 |
| Searching/sharing info | Searching for info on current events | 1 | 5 | 2.20 | 1.637 | 1.280 |
| | Sharing info on current events | 1 | 5 | 1.85 | 1.134 | 1.065 |
| | Uploading personal content/comments | 1 | 5 | 2.16 | 1.661 | 1.289 |
| Engaging with brands/influencers | Searching for info on products, brands, sales | 1 | 5 | 1.76 | 1.049 | 1.024 |
| | Sharing info on products, brands, sales | 1 | 5 | 1.58 | 0.826 | 0.925 |
| | Subscribing to brand fan pages | 1 | 5 | 1.55 | 1.044 | 1.022 |
| | Subscribing to influencers' fan pages | 1 | 5 | 1.78 | 1.485 | 1.219 |
| | Participating in contests | 1 | 5 | 1.62 | 0.983 | 0.991 |

Compiled by the authors.

Frequency data on routine uses was supplemented with information on the types of brands and influencers with which minors seek to interact. For this, the minors surveyed were asked about the following topics:

- Usefulness of social media to reach out to known brands, presented as a dichotomous answer question: Yes (1); No (2) (M = 1.65; $S^2$ = 0.228; SD = 0.478).
- Usefulness of social media to search for information on new brands, presented as a dichotomous answer question: Yes (1); No (2) (M = 1.60; $S^2$ = 0.240; SD = 0.490).
- References by minors surveyed to three brands followed on social media, presented as an open-ended question. All brand references were catalogued and a total of 105 brands were recorded. However, for publishing purposes, only the 20 most repeated brands are listed here.
- References by minors surveyed to three influencers followed on social media, presented as an open-ended question. As for the previous variable, all influencers were recorded (a total of 173) but only the 20 most repeated ones are listed here.

The existence of significant differences between variables with more than two categories (as in the case of the socioeconomic level) was determined by applying the Bonferroni multiple comparisons test, and thus different databases could be analyzed. With the aim of answering the research questions, the results of frequency of use are presented segmented by age, gender, and SES (Socieconomic status) in order to verify to what extent these variables affect the relationship that minors have with brands on social media.

## 4. Results

### 4.1. Frequency of Use of Social Media: Contacting, Sharing and Engaging with Brands

Minors were asked about the frequency with which they perform three activities on social media: contacting, searching for/sharing content with, and engaging with brands/influencers.

The activity most frequently mentioned by minors was contacting close friends or acquaintances; more than half of the sample (54.0%) stated this activity as frequent or very frequent. As can be seen in Table 3 this activity shows the highest average. 7.4% reported no use of social media as a communication tool, a percentage that is significantly higher among the younger cohort (10- to 12-year-olds). In this study, age established a significant difference in the type of routine use. As can be seen in Table 1, the older the child, the greater the tendency to contact with brands.

**Table 3.** Level of frequency with which minors contact through social media. Results are segmented by gender, age, and SES.

| CONTACTING | | Total | Gender | | Age | | SES | | | |
|---|---|---|---|---|---|---|---|---|---|---|
| | | | Boys | Girls | 10 to 12 | 13 to 14 | C1 | C2 | C3 | D |
| Contacting friends/ acquaintances | Never | 7.4% | 6.9% | 7.8% | 9.4% * | 4.5% | 2.8% | 4.3% | 8.4% | 9.3% |
| | Infrequently | 22.2% | 25.1% | 19.6% | 26.8% * | 15.3% | 13.9% | 12.0% | 23.8% | 26.5% * (C2) |
| | Somewhat frequently | 16.4% | 19.0% | 14.1% | 17.4% | 14.9% | 11.1% | 14.1% | 18.9% | 15.8% |
| | Frequently | 25.3% | 20.8% | 29.3% * | 23.4% | 28.2% | 47.2% * (C3, D) | 25.0% | 24.5% | 22.3% |
| | Very frequently | 28.7% | 28.1% | 29.3% | 23.1% | 37.1% * | 25.0% | 44.6% * (C3, D) | 24.5% | 26.0% |

* Statistically significant results between columns within the same variable (gender, age, SES). Results are based on two-sided tests with a significance level of 0.05. Tests are adjusted for all pairwise comparisons within each row within each subtable using the Bonferroni correction. Compiled by the authors.

Regarding contact with brands through social media, the segmentation of the sample by gender also introduces some significant differences. In this case, girls declared contacting friends or acquaintances frequently or very frequently. With reference to socioeconomic status, children in the C1 and C2 segments contact brands more often than other segments.

Table 4 collects frequency data on information searching and sharing on social media. The activity level for this routine is much lower compared to the frequency with which minors establish contact. Approximately between 40% and 50% of the sample stated that they had never shared information, uploaded personal content or comments, nor had they reported on topics of interest. Uploading personal content/leaving comments is more widespread (19.4% do so frequently or very frequently) than searching for information (18.8%) or sharing content (9.0%).

As proved to be the case regarding contacting friends and acquaintances, age is a transcendental variable—the percentage of children in 10- to 12-year-old group who have never performed any of the activities in this second thematic block is significantly higher than in the 13- and 14-year-old cohort. The older children are, the larger the percentage of minors who search for, share and upload information on social media. By gender, boys are more likely to post and share than girls. Segmentation by SES barely showed few and small significant differences.

Compared with the other two activities analyzed involving minors and social media, the results show that interaction with brands and products through social media is the one that generated the least interest among the minors surveyed, as can be stated from analyzing the responses shown in Table 5, and is supported by the averages shown on Table 3. Around 10% of children stated they maintain frequent or very frequent contact with brands through their social media profiles: 8.60% searched for info on products, brands and sales; 11.4% shared information on products or brands, 8.0% subscribed to brand or product fan pages. Within this type of activity, participation in promotional contests showed the lowest percentage (6.8%) which is also lower than the percentage of children who stated frequently following profiles of artists or influencers (14.4%).

However, when results are segmented by age, the percentage of children from the older cohort (13- and 14-year-olds) who stated they maintained a higher frequency of inter-

action with the brands was higher than for the younger members of the study. Regarding gender, girls tend to have a more frequent relationship with brands and products through social media.

When the minors surveyed were asked whether social media had brought them closer to different brands, 35.1% confirmed that they had. Moreover, almost 40% acknowledged having searched for new products and brands through these platforms. Once again, age marked significant differences, a higher percentage of minors in the 13- and 14-year-old group responded affirmatively when asked if social media brought them closer to brands, both new to them (45.0%) and previously known (41.1%).

**Table 4.** Level of frequency with which the minor search for and share content on social media. Results are by segmented by gender, age, and SES.

| SEARCHING AND SHARING INFO | | Total | Gender | | Age | | SES | | | |
|---|---|---|---|---|---|---|---|---|---|---|
| | | | Boys | Girls | 10 to 12 | 13 to 14 | C1 | C2 | C3 | D |
| Searching for info on current events | Never | 39.3% | 40.7% | 38.1% | 47.5% * | 27.2% | 33.3% | 44.6% | 34.3% | 43.3% |
| | Infrequently | 27.7% | 29.4% | 26.3% | 29.1% | 25.7% | 36.1% | 26.1% | 31.5% | 24.2% |
| | Somewhat frequently | 14.2% | 14.3% | 14.1% | 10.7% | 19.3% * | 13.9% | 15.2% | 16.1% | 12.6% |
| | Frequently | 11.0% | 9.5% | 12.2% | 9.4% | 13.4% | 8.3% | 5.4% | 9.1% | 14.0% |
| | Very frequently | 7.8% | 6.1% | 9.3% | 3.3% | 14.4% * | 8.3% | 8.7% | 9.1% | 6.0% |
| Sharing info on current events | Never | 48.9% | 49.8% | 48.1% | 56.5% * | 37.6% | 44.4% | 43.5% | 53.1% | 49.8% |
| | Infrequently | 29.5% | 27.3% | 31.5% | 28.4% | 31.2% | 30.6% | 34.8% | 25.9% | 31.2% |
| | Somewhat frequently | 12.6% | 12.1% | 13.0% | 9.7% | 16.8% * | 16.7% | 12.0% | 12.6% | 11.2% |
| | Frequently | 5.4% | 7.8% * | 3.3% | 3.3% | 8.4% * | 2.8% | 5.4% | 5.6% | 5.1% |
| | Very frequently | 3.6% | 3.0% | 4.1% | 2.0% | 5.9% * | 5.6% | 4.3% | 2.8% | 2.8% |
| Uploading personal content/comments | Never | 42.9% | 43.3% | 42.6% | 50.2% * | 32.2% | 36.1% | 33.7% | 44.1% | 47.9% |
| | Infrequently | 24.8% | 26.0% | 23.7% | 22.1% | 28.7% | 30.6% | 22.8% | 27.3% | 22.3% |
| | Somewhat frequently | 13.0% | 10.4% | 15.2% | 10.0% | 17.3% * | 16.7% | 13.0% | 7.7% | 15.8% |
| | Frequently | 12.4% | 9.5% | 14.8% | 11.4% | 13.9% | 13.9% | 16.3% * (D) | 14.7% | 9.3% |
| | Very frequently | 7.0% | 10.8% * | 3.7% | 6.4% | 7.9% | 2.8% | 14.1% | 6.3% | 4.7% |

* Statistically significant results between columns within the same variable (gender, age, SES). Results are based on two-sided tests with a significance level of 0.05. Tests are adjusted for all pairwise comparisons within each row within each subtable using the Bonferroni correction. Compiled by the authors.

**Table 5.** Level of frequency with which minors interact with brands on social media. Results are segmented by gender, age, and SES.

| RELATIONSHIP WITH BRANDS/INFLUENCERS | | Total | Gender | | Age | | SES | | | |
|---|---|---|---|---|---|---|---|---|---|---|
| | | | Boys | Girls | 10 to 12 | 13 to 14 | C1 | C2 | C3 | D |
| Searching for info on products, brands, sales | Never | 53.7% | 53.2% | 54.1% | 62.2% * | 41.1% | 50.0% | 40.2% | 56.6% | 58.6% (C2) |
| | Infrequently | 28.1% | 30.7% | 25.9% | 24.4% | 33.7% * | 33.3% | 37.0% | 28.7% | 23.3% |
| | Somewhat frequently | 9.6% | 9.5% | 9.6% | 8.4% | 11.4% | 2.8% | 10.9% | 9.1% | 9.8% |
| | Frequently | 6.0% | 4.3% | 7.4% | 3.7% | 9.4% * | 11.1% | 7.6% | 4.2% | 5.6% |
| | Very frequently | 2.6% | 2.2% | 3.0% | 1.3% | 4.5% * | 2.8% | 4.3% | 1.4% | 2.8% |
| Sharing info on products, brands and sales | Never | 62.9% | 59.7% | 65.6% | 69.9% * | 52.5% | 47.2% | 44.6% | 65.0% * (C2) | 72.1% * (C1, C2) |
| | Infrequently | 24.2% | 22.9% | 25.2% | 21.4% | 28.2% | 41.7% (D) | 38.0% (D) | 23.8% | 15.8% |
| | Somewhat frequently | 6.6% | 9.1% * | 4.4% | 4.0% | 10.4% * | 8.3% | 9.8% | 7.0% | 4.2% |
| | Frequently | 4.8% | 5.6% | 4.1% | 3.7% | 6.4% | 0.0% | 6.5% | 3.5% | 5.6% |
| | Very frequently | 1.6% | 2.6% | 0.7% | 1.0% | 2.5% * | 2.8% | 1.1% | 0.7% | 2.3% |

**Table 5.** *Cont.*

| RELATIONSHIP WITH BRANDS/INFLUENCERS | | Total | Gender | | Age | | SES | | | |
|---|---|---|---|---|---|---|---|---|---|---|
| | | | Boys | Girls | 10 to 12 | 13 to 14 | C1 | C2 | C3 | D |
| Subscribing to brand fan pages | Never | 70.7% | 71.4% | 70.0% | 76.9% * | 61.4% | 75.0% | 67.4% | 65.7% | 74.0% |
| | Infrequently | 15.2% | 15.6% | 14.8% | 12.7% | 18.8% | 13.9% | 17.4% | 17.5% | 14.0% |
| | Something frequently | 6.2% | 5.2% | 7.0% | 4.7% | 8.4% | 8.3% | 4.3% | 6.3% | 6.0% |
| | Frequently | 4.8% | 4.8% | 4.8% | 3.3% | 6.9% | 2.8% | 5.4% | 7.0% | 3.7% |
| | Very frequently | 3.2% | 3.0% | 3.3% | 2.3% | 4.5% | 0.0% | 5.4% | 3.5% | 2.3% |
| Suscribing to influencers' fan pages | Never | 62.9% | 66.7% | 59.6% | 68.2% * | 55.0% | 63.9% | 67.4% | 59.4% | 64.2% |
| | Infrequently | 15.2% | 15.2% | 15.2% | 14.4% | 16.3% | 16.7% | 13.0% | 18.2% | 14.0% |
| | Somewhat frequently | 7.6% | 7.8% | 7.4% | 8.4% | 6.4% | 8.3% | 5.4% | 7.7% | 8.4% |
| | Frequently | 9.4% | 7.8% | 10.7% | 6.0% | 14.4% * | 5.6% | 10.9% | 9.1% | 8.8% |
| | Very frequently | 5.0% | 2.6% | 7.0% * | 3.0% | 7.9% * | 5.6% | 3.3% | 5.6% | 4.7% |
| Participating in promotional contests | Never | 61.7% | 60.6% | 62.6% | 65.9% * | 55.4% | 61.1% | 60.9% | 60.8% | 63.7% |
| | Infrequently | 24.2% | 25.5% | 23.0% | 24.4% | 23.8% | 22.2% | 26.1% | 23.1% | 24.7% |
| | Somewhat frequently | 7.4% | 6.9% | 7.8% | 4.7% | 11.4% * | 11.1% | 7.6% | 5.6% | 6.0% |
| | Frequently | 3.6% | 2.2% | 4.8% | 2.3% | 5.4% | 5.6% | 4.3% | 5.6% | 1.9% |
| | Very frequently | 3.2% | 4.8% | 1.9% | 2.7% | 4.0% | 0.0% | 1.1% | 4.9% | 3.7% |

\* Statistically significant results between columns within the same variable (gender, age, SES). Results are based on two-sided tests with a significance level of 0.05. Tests are adjusted for all pairwise comparisons within each row within each subtable using the Bonferroni correction. Compiled by the authors.

### 4.2. Main Brands and Influencers Followed on Social Media

The study also recorded the type of products and brands that generate the most interest among children on social media. As can be seen in Figure 1, sportswear (46%), and clothing and fashion (22%), are the most popular types of brands. To a lesser extent, minors also showed an inclination towards brands in the areas of technology (9.0%), food/beverages, toys, and video and music services (4%).

Most of the minors acknowledged following one brand on social media, only 4.2% of those who are registered on social media asserted not doing so. Table 6 shows the 20 most cited brands in the questionnaire by children. As for sports brands, two brands, Adidas and Nike, are the top of the list. It is also interesting to note that among the list of most widely recalled brands, minors include two of the main Chilean retail stores (Falabella and Paris). Furthermore, the complete list includes other retail stores (Ripley, La Polar). Likewise, by technology category, minors have included the names of social media, such as YouTube, as well as brands belonging to the mobile telephone and video game industries. Coca-Cola is also among the most followed brands on social media, as well as Uber and Mac Cosmetics, although to a lesser extent.

Sportswear brands such as Adidas, Nike, or Jordan were mentioned more by boys than by girls, unlike Falabella, which was mainly followed by girls. By age, Vans is more related to the older cohort (13- and 14-year-olds). By socioeconomic strata, minors belonging to the C1 level were more interested in retail companies such as Falabella and Paris than those in the D level.

Given the role that influencers play in the digital routines of minors, it is relevant to know which of them are followed by minors on their social media. As can be seen in Table 7, practically one third of the children surveyed (31.6%) stated that they did not follow any influencers.

Children mostly follow singers (reggaeton and K-pop), soccer players, and Spanish or English-speaking YouTubers. By age, gender, and SES, most statistically significant differences are observed regarding gender. Most of the YouTubers followed by minors stand out for being gamers (HolaSoyGermán, ElrubiusOMG, Fernafloo, and Ninja). By gender, the biggest difference involves preferences over soccer players, who are mainly followed by boys. Girls, on the other hand, mainly seek to connect with Latino singers and South Korean bands. Regarding the SES, although the Bonferroni test was unable

to assign statistical significance to the difference, more than 50% of the minors in the D stratum indicated that they did not follow influencers on social media. Interestingly, the influencers most followed by members of the C1 status are least present in the list that summarizes the top 20 influencers (with the exception of Chilean soccer player Alexis Sánchez). C3 status members seem to follow influencers the most (only 2.5% said they followed none), especially YouTubers. In reference to the segmentation by age, differences are hardly present.

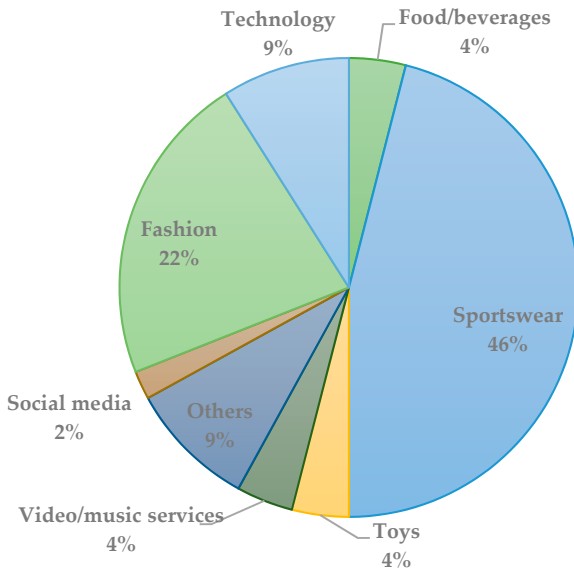

**Figure 1.** Main categories of products and brands followed by minors on social media. Compiled by the authors.

**Table 6.** List of the 20 brands most frequently recalled by children on social media. Results segmented by gender, age and SES.

| Category | Brand | Total | Gender | | Age | | SES | | | |
|---|---|---|---|---|---|---|---|---|---|---|
| | | | Boys | Girls | 10 to 12 | 13 to 14 | C1 | C2 | C3 | D |
| **Sportwear** | Adidas | 13.2% | 16.5% * | 10.4% | 12.7% | 13.9% | 11.1% | 15.2% | 12.6% | 13.5% |
| | Nike | 11.8% | 16.5% * | 7.8% | 10.7% | 13.4% | 5.6% | 13.0% | 13.3% | 11.6% |
| | Puma | 2.2% | 2.6% | 1.9% | 2.3% | 2.0% | 0.0% | 2.2% | 2.8% | 2.3% |
| | Jordan | 2.0% | 3.5% * | 0.7% | 2.3% | 1.5% | 0.0% | 0.0% | 4.2% | 1.9% |
| | Everlast | 0.8% | 0.4% | 1.1% | 0.7% | 1.0% | 0.0% | 1.1% | 0.0% | 1.4% |
| **Fashion/clothes** | Converse | 2.2% | 1.3% | 3.0% | 2.3% | 2.0% | 2.8% | 1.1% | 3.5% | 1.4% |
| | Vans | 1.6% | 1.3% | 1.9% | 0.7% | 3.0% * | 0.0% | 2.2% | 1.4% | 1.4% |
| | Gucci | 1.2% | 1.7% | 0.7% | 1.0% | 1.5% | 0.0% | 2.2% | 1.4% | 0.9% |
| | Supreme | 0.8% | 1.3% | 0.4% | 0.7% | 1.0% | 0.0% | 0.0% | 1.4% | 0.9% |
| **Retail stores** | Falabella | 1.8% | 0.4% | 3.0% * | 2.3% | 1.0% | 8.3% * (D) | 1.1% | 2.8% | 0.5% |
| | Paris | 1.8% | 0.9% | 2.6% | 2.3% | 1.0% | 8.3% * (D) | 2.2% | 1.4% | 0.9% |
| | Ripley | 0.6% | 0.4% | 0.7% | 0.7% | 0.5% | 5.6% | 1.1% | 0.0% | 0.0% |
| **Technology** | YouTube | 1.6% | 1.3% | 1.9% | 2.3% | 0.5% | 0.0% | 3.3% | 1.4% | 0.9% |
| | Samsung | 1.2% | 1.7% | 0.7% | 1.0% | 1.5% | 0.0% | 0.0% | 2.1% | 0.9% |
| | Facebook | 0.8% | 0.9% | 0.7% | 0.7% | 1.0% | 0.0% | 2.2% | 0.0% | 0.9% |
| | LG | 0.8% | 0.9% | 0.7% | 0.3% | 1.5% | 0.0% | 0.0% | 0.7% | 1.4% |
| | PlayStation | 0.6% | 1.3% | 0.0% | 0.0% | 1.5% | 0.0% | 1.1% | 0.7% | 0.0% |
| **Others** | Coca-Cola | 1.6% | 1.3% | 1.9% | 1.7% | 1.5% | 2.8% | 1.1% | 2.1% | 1.4% |
| | Uber | 1.0% | 1.3% | 0.7% | 0.3% | 2.0% | 0.0% | 0.0% | 1.4% | 1.4% |
| | Mac Cosmetics | 0.6% | 0.0% | 1.1% | 0.3% | 1.0% | 2.8% | 0.0% | 0.7% | 0.5% |
| **No brands followed on SM** | | 4.2% | 3.0% | 5.2% | 5.0% | 3.0% | 2.8% | 4.3% | 3.5% | 4.7% |

\* Statistically significant results between columns within the same variable (gender, age, SES). Results are based on two-sided tests with a significance level of 0.05. Tests are adjusted for all pairwise comparisons within each row within each subtable using the Bonferroni correction. Compiled by the authors.

**Table 7.** Categorized list of the influencers most followed by the analyzed sample on social networks. Results are segmented by gender, age and SES.

| Category | Influencer | Total | Gender | | Age | | SES | | | |
|---|---|---|---|---|---|---|---|---|---|---|
| | | | Boys | Girls | 10 to 12 | 13 to 14 | C1 | C2 | C3 | D |
| Singers | Nacho | 13.4% | 18.1% | 9.9% | 16.9% | 8.1% | 15.4% | 7.0% | 11.4% | 17.0% |
| | Bad Bunny | 9.3% | 8.6% | 9.9% | 10.8% | 7.1% | 7.7% | 2.3% | 8.9% | 12.3% |
| | BTS | 6.1% | 0.0% | 10.6% * | 4.1% | 9.1% | 0.0% | 7.0% | 3.8% | 6.6% |
| | Katie Ángel | 5.3% | 0.0% | 9.2% * | 6.8% | 3.0% | 0.0% | 4.7% | 8.9% | 2.8% |
| | Ozuna | 4.5% | 3.8% | 4.9% | 5.4% | 3.0% | 0.0% | 0.0% | 3.8% | 7.5% |
| | BLACKPINK | 4.5% | 1.0% | 7.0% * | 3.4% | 6.1% | 0.0% | 4.7% | 3.8% | 4.7% |
| | Kimberly Loaiza | 4.5% | 0.0% | 7.7% * | 4.7% | 4.0% | 0.0% | 2.3% | 8.9% | 1.9% |
| | CNCO | 4.0% | 0.0% | 7.0% * | 6.8% * | 0.0% | 0.0% | 0.0% | 7.6% | 3.8% |
| | Maluma | 3.6% | 1.9% | 4.9% | 4.7% | 2.0% | 0.0% | 0.0% | 6.3% | 3.8% |
| | Sofía Castro | 3.2% | 0.0% | 5.6% * | 4.1% | 2.0% | 0.0% | 2.3% | 3.8% | 3.8% |
| | Los Polinesios | 3.2% | 0.0% | 5.6% * | 4.7% | 1.0% | 0.0% | 2.3% | 5.1% | 2.8% |
| | Juan de Dios Pantoja | 3.2% | 1.0% | 4.9% | 4.1% | 2.0% | 0.0% | 2.3% | 3.8% | 2.8% |
| Athletes (footballers) | Cristiano Ronaldo | 6.9% | 16.2% * | 0.0% | 5.4% | 9.1% | 0.0% | 9.3% | 8.9% | 5.7% |
| | Arturo Vidal | 6.1% | 11.4% * | 2.1% | 5.4% | 7.1% | 7.7% | 11.6% | 6.3% | 3.8% |
| | Alexis Sánchez | 5.7% | 10.5% * | 2.1% | 2.7% | 10.1% | 30.8% * (C3, D) | 7.0% | 3.8% | 3.8% |
| | Messi | 5.3% | 12.4% * | 0.0% | 4.7% | 6.1% | 0.0% | 7.0% | 7.6% | 3.8% |
| YouTubers/ gamers | HolaSoyGermán | 10.9% | 15.2% | 7.7% | 11.5% | 10.1% | 0.0% | 7.0% | 17.7% | 7.5% |
| | ElrubiusOMG | 4.5% | 7.6% | 2.1% | 4.1% | 5.1% | 0.0% | 0.0% | 5.1% | 5.7% |
| | Fernafloo | 4.5% | 8.6% * | 1.4% | 4.7% | 4.0% | 0.0% | 0.0% | 6.3% | 3.8% |
| | Ninja | 3.2% | 6.7% * | 0.7% | 2.0% | 5.1% | 0.0% | 4.7% | 5.1% | 1.9% |
| **No influencers followed on SM** | | 31.6% | 30.5% | 32.4% | 37.8% | 22.2% | 38.5% | 30.2% | 2.5% | 54.7% |

\* Statistically significant results between columns within the same variable (gender, age, SES). Results are based on two-sided tests with a significance level of 0.05. Tests are adjusted for all pairwise comparisons within each row within each subtable using the Bonferroni correction. Compiled by the authors.

## 5. Discussion

This study shows that the main reason for which minors use social media is to contact friends or acquaintances, rather than to maintain a relationship with brands or influencers.

However, minors are not immune to the influence of social media. The involvement of minors in social media increases as they grow older, particularly regarding their connection with influencers. Within the activities categorized as engaging with brands, connecting with influencers is the most recurrent within the analyzed age group.

Advertisers present on social media seek to generate their own communication channel and establish a closer and two-way relationship with their audience. Although access to these communication channels is for free, the information that flows through them is not intention-free, a fact which minors and parents must always bear in mind.

It must be noted that the brands most followed by minors on social media belong to the industries which have successfully endorsed e-commerce in Chile. In 2019, the Chamber of Commerce of Santiago stated that e-commerce had intensively penetrated the electronics, mobile phone, video, and computing markets. Purchases in these markets are heavily influenced by the information people gather and by personal recommendations, correspond to products in which children have special purchase decision [35]. The fashion industry has grown rapidly in recent years and leads in number of transactions, while e-commerce in the food industry is starting to take off in Chile, and represented the fastest growing area in 2019 [35]. Likewise, many of the brands minors can recall correspond to those that spend the most money on advertising in the Andean country, as is the case of large department stores, such as Falabella and Paris, who lead in advertising investment in conventional media [36].

Although there is some interaction on the part of minors with brands through official channels, what seems to most generate their interest is searching for information or exchanging comments with influencers; 14.4% declared they subscribed to fan pages of influencers frequently or very frequently, a percentage that increases to 22% if only the older children surveyed are considered. In addition, when the statistics are analyzed as a whole, brands show higher dispersion (with the exception of Adidas and Nike) compared to that of names of influencers. The latter tend to revolve around names in three sectors: the music industry, soccer, and YouTube. These results are in line with those of previous studies which had identified the admiration children feel toward influencers and their aspiration to become YouTubers [37].

Brands reach out to young audiences online by offering them immersive, entertaining content and/or the presence of celebrities [38]. Indeed, this study highlights the fact that celebrities attract the attention of minors, to the point that minors themselves describe music artists and soccer players who are not particularly expert in the area as influencers. This proves to be especially true for the socioeconomic levels C3 and D. The closeness minors seem to feel toward YouTubers may derive from the levels of knowledge they have in certain areas of interest to minors, such as video games. Therefore, it seems necessary to continue research into the features that lead minors to label certain opinion leaders as influencers, and thus determine which elements are associated with credibility by children during their selection process. Finally, the fact minors seem to focus their use of social media on communicative purposes could raise new questions about the extent to which this social use is associated with consumer issues (talking about influencers or brands).

It would seem that influencer marketing on social media is becoming an increasingly attractive way for advertisers to interact with new generations in a less intrusive way. The results of this study contribute to those of previous studies on influencer marketing addressed at children and which have shown that adolescents accept the presence of brands and sponsorships in the contents disclosed by the influencers of their choice, as long as the balance entertainment/commercial content remains undisturbed [13,14]. Given that influencers generate interest and are followed through social media, this leads brands to want to count them in when implementing new ways of advertising [39].

Now, faced with this phenomenon, it is worth asking about the ability minors have to identify the persuasive intentionality of the messages they receive from the influencers they follow on social networks, that is, their level of advertising literacy [40] especially in the face of content that increasingly mixes advertising and entertainment [41,42]. Previous studies have researched the recognition of advertising on social networks by minors and have concluded that these generally identify standard formats as advertising, but more camouflaged commercial messages are not considered as such (unboxing or recommendations from YouTubers and Instagrammers) [43]. This fact, therefore, truncates the first step in advertising literacy, which is the recognition of the phenomenon [44]. Hence it is necessary to work with minors and have them develop their ability to question whether the publications of the influencers they follow on social networks may be sponsored content.

The relationship between minors and brands tends to grow as children grow older, and they start searching for information, sharing content, or following their official webpages. It must be noted that, even if minimally, brands are also contacting an audience that is aged 10 to 12 and which theoretically should not have access to social media, given that legally minors must be 13 or 14 years old—depending on the platform—as well as have parental consent. This opens new lines of debate and research on the type of persuasive content to which minors are exposed to through brands and influencers on social media. The suitability of the messages with which brands target an audience that has an appealing future consumer role needs to be assessed, particularly given that these children are still in the process of developing intellectually. As brought to light by this study, age determined significant differences in the frequency of use of social media, and gender determined choice of brands and influencers. Thus, while boys chose to follow sportswear brands and soccer players and YouTubers, girls showed a greater interest in clothing stores and brands

and in following music artists. There seems to be evidence to support the notion that minors maintain a similar consumption pattern in terms of gender, regardless of the screen or service [45]. Regarding the segmentation of the results according to the socioeconomic level of the minor's household, although no particularly significant differences were observed, a greater preference for the profiles of prominent influencers was perceived in levels C3 and D, a trend that merits further research.

In short, it can be concluded that minors aged 10 to 14 years old maintain some kind of contact with brands through social media, especially related to sports, fashion, and technology sectors. Establishing contact with minors through influencers, mainly music artists, soccer players and YouTubers, becomes an attractive method of communication with younger generations, which does not go unnoticed by brands.

The quantitative and exploratory approach of this study limits its scope. However, there is a need to continue conducting future qualitative research through questionnaires that enable children to express themselves openly and reveal their attitude and predisposition toward the advertisers and influencers with whom they interact on social media. For instance, and taking into consideration the links that brands and minors establish, it would be relevant to learn whether these bonds lead children to consume specific products or services directly or indirectly later on in life.

For all of the above, the challenge continues to be to provide minors with comprehensive advertising literacy that will allow them to enrich their dialogs with brands, and to recognize and interpret the types of messages and content presented by these platforms whose main goal is to turn interest into online sales.

## 6. Conclusions

The results of this research show that the interaction of minors with brands and influencers through social networks is a relatively important activity in the digital routines of the age group studied (10–14 years). Specifically, the interests minors have and the type of advertisers and opinion leaders with whom they are willing to bond through these platforms are defined. Brands that attract their attention the most are sports, fashion, and technology brands. However, minors expressed a greater preference for following celebrities than specific brands on social networks, mainly from the world of music and football, and YouTubers, which makes influencer marketing gain strength among young consumers.

Consequently, in these circumstances in which nothing is what it seems and the differentiation of the different types of content is increasingly complex, an advertising literacy that promotes skeptical and questioning attitudes towards the content that is received through social networks becomes increasingly necessary. Therefore, it is suggested that researchers continue gaining understanding of the relationship minors have with brands and influencers on these platforms in order to promote healthy and sustainable interactions between advertisers and future adults. Ultimately, we seek to have children be able to immerse themselves in the consumption of content on social networks with a patient and critical look towards the media and digital advertising.

**Author Contributions:** B.F. has carried out the funding acquisition, the methodology, the validation, the formal analysis, the investigation, the resources and data curation. C.S. has developed the theorical framework and the writing—original draft preparation. Both authors have read and agreed to the published version of the manuscript.

**Funding:** This research was funded by THE NATIONAL COMMISSION FOR SCIENTIFIC AND TECHNOLOGICAL RESEARCH (CONICYT) OF THE GOVERNMENT OF CHILE, grant number 11170336, Fondecyt Initiation project.

**Institutional Review Board Statement:** The study was conducted according to the guidelines of the Declaration of Helsinki, and approved by the Ethics Committee of La Universidad de los Andes, Chile (CE 13/18).

**Informed Consent Statement:** Informed consent was obtained from all subjects involved in the study.

**Data Availability Statement:** The data presented in this study are available on request from the corresponding author.

**Conflicts of Interest:** The authors declare no conflict of interest.

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
