# Peer review of "The Relationship of Chilean Minors with Brands and Influencers on Social Networks"

_sustainability, doi:10.3390/su13052822_

Round 1
Reviewer 1 Report
The results of this paper prove that minor-brand interaction through social media and influencers is an activity that has significant importance at the age bracket studied (10-14 years) in Chile. The results of this research project suggest that this activity should be more closely examined in order to promote healthy and sustainable relationships between brands and future adults.
The research is well done and the article is well written. Since it is an exploratory quantitative descriptive research, you only hint at the potential dangers or opportunities of those interactions. I hope that those will be explored in future works as it is suggested in the discussion of the paper. Advertising literacy seems to be a pressing issue for Chilean youth.
Differences between age brackets and socioeconomic status seem also relevant and interesting, and they should receive more attention from future studies: Is there an economic breach in the youth's relationship with brands and influencers? Which is the right age to start those "advertising literacy" programs?
Reviewer 2 Report
The theoretical framework is in accordance with the study addressed. The authors manage to synthesize the most relevant in an optimal space. In the methodological part, reference is made to a census of the year 2017. We are in the year 2021, perhaps it would be convenient to update the data. Perhaps, in the methodology it is could describe something else why quantitative analysis is considered interesting: what it contributes, etc. I do not know if in the country where the study is carried out there are ethical standards (when doing a study on minors) that should be reflected at some point in the study. In other matters, I would try to use different colors on graphics as figure 1.Accept after minor revision
Reviewer 3 Report
Comments
The article presents the results of their study that attempted to analyse the relationship between two groups of children (aged 10-12 and 13-14) and brands on social media.
Literature review:
- The authors of the article should have considered that the cognitive development of this two-age group is quite different, and this fact only would influence their results. The literature review should include at least one paragraph on this issue.
- The topic of influencer marketing to children is also very sparsely mentioned, not to the mention the literature of brand awareness and brand recognition.
- Ethical aspects of advertising to children and adolescents. The authors neglect this important area. Children have become the key target for many advertisers. Children are vulnerable, easy to exploit consumers and they perceive things as advertisers want them to perceive or believe. Even though children are nowadays smart and knowledgeable of the marketplace nevertheless for many marketers they are relatively easy to target due to the sheer size of the children’s consumer market with growing financial power.
Research:
The authors do not give an appropriate description about how and by whom the face-to-face interview of the minors were carried out? Who conducted the interviews? Did the interviewers had any training on how to conduct interviews with children at this age? Did they have permissions of the parents? Were the parents present during the interviews?
Discussion and conclusions
This section is way too short and only summarising the results. Please rewrite this section and give insight why this research was useful!
Round 2
Reviewer 3 Report
The authors took the revision seriously and revised well the points I mentioned in my review.